# Is High Expression of Claudin-7 in Advanced Colorectal Carcinoma Associated with a Poor Survival Rate? A Comparative Statistical and Artificial Intelligence Study

**DOI:** 10.3390/cancers14122915

**Published:** 2022-06-13

**Authors:** Victor Ianole, Mihai Danciu, Constantin Volovat, Cipriana Stefanescu, Paul-Corneliu Herghelegiu, Florin Leon, Adrian Iftene, Ciprian-Gabriel Cusmuliuc, Bogdan Toma, Vasile Drug, Delia Gabriela Ciobanu Apostol

**Affiliations:** 1Pathology Department, Grigore T. Popa University of Medicine and Pharmacy Iasi, 700115 Iasi, Romania; ianole.victor@umfiasi.ro (V.I.); bogdan-toma@students.umfiasi.ro (B.T.); delia.ciobanu@umfiasi.ro (D.G.C.A.); 2Sf. Spiridon Emergency Clinical Hospital Iasi, 700111 Iasi, Romania; 3Department of Medical Oncology, Grigore T. Popa University of Medicine and Pharmacy/Euroclinic Oncology Center Iasi, 700115 Iasi, Romania; volovat.constantin@umfiasi.ro; 4Nuclear Medicine Laboratory, Grigore T. Popa University of Medicine and Pharmacy Iasi, 700115 Iasi, Romania; cipriana.stefanescu@umfiasi.ro; 5Faculty of Automatic Control and Computer Engineering, Gheorghe Asachi Technical University of Iasi, 7000050 Iasi, Romania; florin.leon@academic.tuiasi.ro; 6Faculty of Computer Science, Alexandru Ioan Cuza University of Iasi, 700259 Iasi, Romania; adiftene@info.uaic.ro (A.I.); gabriel.cusmuliuc@info.uaic.ro (C.-G.C.); 7Gastroenterology Department, Grigore T. Popa University of Medicine and Pharmacy Iasi, 700115 Iasi, Romania; vasile.drug@umfiasi.ro

**Keywords:** Claudin-7, advanced colorectal carcinoma, cancer, prognostic biomarkers, artificial intelligence models, immunohistochemistry

## Abstract

**Simple Summary:**

The need for predictive and prognostic biomarkers in colorectal carcinoma (CRC) brought us to an era where the use of artificial intelligence (AI) models is increasing. We investigated the expression of Claudin-7, a tight junction component, which plays a crucial role in maintaining the integrity of normal epithelial mucosa, and its potential prognostic role in advanced CRCs by drawing a parallel between statistical and AI algorithms. Claudin-7 immunohistochemical expression was evaluated in the tumor core and invasion front of CRCs and correlated with clinicopathological parameters and survival using statistical and AI algorithms. The Kaplan–Meier univariate survival analysis showed that the immunohistochemical overexpression of Claudin-7 in the tumor invasive front may represent a poor prognostic factor in advanced stages of CRCs. On the contrary, AI models could not predict the same outcome, probably because of the small number of patients included in our cohort.

**Abstract:**

Aim: The need for predictive and prognostic biomarkers in colorectal carcinoma (CRC) brought us to an era where the use of artificial intelligence (AI) models is increasing. We investigated the expression of Claudin-7, a tight junction component, which plays a crucial role in maintaining the integrity of normal epithelial mucosa, and its potential prognostic role in advanced CRCs, by drawing a parallel between statistical and AI algorithms. Methods: Claudin-7 immunohistochemical expression was evaluated in the tumor core and invasion front of CRCs from 84 patients and correlated with clinicopathological parameters and survival. The results were compared with those obtained by using various AI algorithms. Results: the Kaplan–Meier univariate survival analysis showed a significant correlation between survival and Claudin-7 intensity in the invasive front (*p* = 0.00), a higher expression being associated with a worse prognosis, while Claudin-7 intensity in the tumor core had no impact on survival. In contrast, AI models could not predict the same outcome on survival. Conclusion: The study showed through statistical means that the immunohistochemical overexpression of Claudin-7 in the tumor invasive front may represent a poor prognostic factor in advanced stages of CRCs, contrary to AI models which could not predict the same outcome, probably because of the small number of patients included in our cohort.

## 1. Introduction

Patients with advanced stages of colorectal carcinoma (CRC) have a high risk of recurrence, with tumors in these stages exhibiting accelerated proliferation, increased tendency towards invasion and metastasis, and heterogeneity in treatment response [1,2,3,4]. Claudin-7, a tight junction component, is one of the most important members of the claudin family, consisting of 211 amino acid residues, which plays a crucial role in maintaining tight junction integrity, epithelial cell polarity, and ion permeability between cells (the integrity of epithelial mucosa) [5,6]. Recently, Claudin-7 has been reported to be also involved in non-tight junction-related functions such as inflammation initiation and in different tumor development steps [7]. Abnormal Claudin-7 expression was reported in a variety of cancers (e.g., ovary, breast, prostate, esophagus, stomach, colon, and lung) and has been associated with tumorigenesis, progression, and metastasis [5,6]. In CRC, up-regulation, down-regulation and even deletion of Claudin-7 have been reported, and are an important step in tumorigenesis, invasion, epithelial-to-mesenchymal transition (EMT), metastasis and even tumor suppression [5,8,9,10].

Although Claudin-7 is involved in the pathogenesis of CRC through several distinct mechanisms, the literature data are inconsistent and limited [10,11,12,13]. Therefore, the aim of the present study was to investigate the expression of Claudin-7 and the potential prognostic role in advanced stages of CRC. Although Artificial Intelligence (AI) is a well-established field of research, in recent years, it has been applied extensively in medicine, mainly using its subdomains, Machine Learning and Deep Learning. In medicine, AI algorithms are commonly used to predict the outcome of a treatment, find correlations among data, or diagnose certain pathologies. Related to CRC, AI algorithms were used to diagnose or foresee the evolution of the disease [14,15] or the outcome of a CRC treatment [16,17] at an early stage. More specifically, AI algorithms were used to diagnose CRC during colonoscopy [18,19], on biopsies [20] or using RNA molecular biology [21]. AI was also widely used in CRC surgery [22,23] or chemotherapy [24]. To our knowledge, no prior study investigating the role of Claudin-7 expression in CRC using AI algorithms has been made.

## 2. Materials and Methods

### 2.1. Study Group

The study group consisted of 84 patients with histologically confirmed advanced stage CRC (stage IV), diagnosed between 2008 and 2020, in “Sf. Spiridon” Emergency County Hospital Iasi, Romania. This study was approved by the Ethics Committee of “Sf. Spiridon” Emergency County Hospital, Iasi and written informed consent was obtained from all patients. 

### 2.2. Tissue Microarray and Immunohistochemistry

Tumor samples were routinely processed by fixation in neutral buffered formalin 10% and paraffin-embedding. Tissue Microarrays (TMAs) were constructed using 2 punches (4 mm diameter) from each case, one from the tumor core and the other from the invasion tumor front. The control group included 20 samples of normal colonic mucosa resection margins harvested from at least 10 cm from adenocarcinoma. Immunohistochemical tests were performed using anti-Claudin-7 monoclonal antibody (rabbit anti-human, 1:1500, ab207300, Abcam, Cambridge, UK) after pretreatment with a specific epitope retrieval solution (pH 9) at 96 °C, for 25 min. The detection of the immunoreaction was performed using an UltraVision LP Detection System (ThermoFisher Scientific, Fremont, CA, USA) and 3,3′-Diaminobenzidine chromogen (DAB) (ThermoFisher Scientific, Fremont, CA, USA). The specificity of the immunoreactivity was checked by omitting the primary antibody and replacing it with non-immunized serum at the same dilution (negative control) for each TMA. The immunohistochemistry tests were performed on two replicates from each TMA.

### 2.3. Immunohistochemical Assessment Protocol

Assessment of Claudin-7 expression was carried out by two independent investigators, in a blind manner. Expression of Claudin-7 was defined as the presence of membranous staining in tumor cells. A semi-quantitative four-tiered scoring system was used to measure the proportion of stained tumor cells per core, as follows: 0 = <5% positive tumor cells; 1 = 5–30%; 2 = 30–60%; and 3 = >60%. The intensity of immunoreaction was evaluated using a four-tiered system: 0—negative, 1—weak, 2—moderate, 3—strong. Both the tumor core and the invasive front were evaluated. Discordant cases were re-evaluated in the panel in order to achieve a consensus.

### 2.4. Statistical Analysis

Data were analyzed using Microsoft Office Excel and IBM Statistical Package for the Social Sciences (SPSS) version 26. The chi-squared test and Fisher’s exact test were used to assess the relationship between Claudin-7 expression and clinicopathological parameters (according to the 8th edition of AJCC Cancer Staging Manual, 2017 and the 5th Edition of WHO Classification of Tumors: Digestive System Tumours, 2019: age, sex, T stage, N stage, grading, tumor location, venous invasion, lymphovascular invasion, perineural invasion, tumor growth pattern, tumor deposits, tumor budding, leukocyte infiltrate) [25,26]. Survival was defined as the time elapsed between the date of diagnosis and therapy initiation, to the date of death or of the last follow-up. For univariate survival, the Kaplan–Meier method and scatter plots were used to estimate the overall survival (OS) and analyzed using the log-rank test. Cox multiple regression was used to perform multivariate survival analysis. A *p*-value < 0.05 was considered to be statistically significant. 

### 2.5. AI Analysis

From the dataset, only 6 Claudin-7 attributes were first selected as inputs: Cldn7 (Core) percentage (P), Cldn7 (Core) intensity (I), Cldn7 (Core) membranous staining pattern: discontinuous vs. continuous, Cldn7 (Front) P, Cldn7 (Front) I, and Cldn7 (Front) membranous staining pattern: discontinuous vs. continuous.

The survival rate was considered as the output, in two different forms, as a classification and as a regression problem. The data were preprocessed to compute the difference between the Visit Date and the Date of Death (when it was the case) in years. Then, three Survival class values were constructed: “2” when the difference was less than 2 years, “5” when it was between 2 and 5 years, and “T” (10) when it was greater than 10 years. When considering the problem in its regression form, the actual difference in years was the output.

Before applying the machine learning algorithms, it is useful to have an overall visual representation of the distribution of the data. In this section, each attribute is considered independently in a predictive relation to the output, which is an oversimplifying assumption, but it can provide some initial information about the problem.

Figure 1 displays this for the classification problem (with 3 classes). One can see that the class values are equally distributed for the different input values, and there is no discernable pattern in this representation.

The same can be said about the numeric values in Figure 2: for each input value (on the abscise) there is a large range of output values (on the ordinate).

Therefore, it cannot be concluded that any input attribute can independently influence the survival rate of the patients.

### 2.6. Scenarios for AI Experiments

The experiments were performed in three scenarios:On the whole training set, to assess the ability of an algorithm to learn the data at all;With 10-fold cross-validation, which is the de facto standard of comparing different algorithms and assessing their generalization capabilities;With the leave-one-out approach, which is useful when the number of training instances is small (in our case, there are 84 patients).


The following algorithms were applied, using the Weka software:
Nearest neighbor (NN);K-nearest neighbor (kNN), with k neighbors and a distance weighting function;Non-nested generalized exemplars (NNGE);C4.5 decision tree;Random forest;Support vector machine (SVM) classification and regression, with a specified kernel, e.g., radial basis function (RBF);Linear models for classification and regression.

The same one-to-one correspondence as carried out in the statistical approach between clinicopathological parameters (as inputs) and Claudin-7 expression (as outputs) was evaluated by means of information gain. This method assesses the relevance of an attribute for solving a classification problem. More specifically, if the class values can be perfectly distinguished by testing the values of an attribute, then that attribute can solve the classification problem by itself. In this ideal case, each attribute value corresponds to a partition of the dataset where the instances have the same class value. The resulting partitions have zero entropy, and thus the difference in entropy between the original dataset and the partitioned dataset is maximum. When the attribute values cannot partition the dataset perfectly, more homogenous partitions are still preferred, and entropy can still be used as a homogeneity measure.

For a class *C* and an attribute *A*, this increase in entropy is defined as:(1)$$InfoGain\left(C,A\right)=H\left(C\right)-H(C|A)$$
where *H* is the entropy:
(2)$$H\left(C\right)=-\overset{c}{\underset{i=1}{\sum }}p(i|C)\log_{2}p(i|C)$$

The probabilities *p* can be directly computed from the data as the proportion of instances with a class value *i.* The same approach is used to compute the conditional entropy for each attribute value *Aj*. Basically, a larger information gain means that an attribute is more relevant for the classification problem.

For the information gain algorithm, we considered ages ranging from 20 to 90 years old as discretized classes for every 10 years.

### 2.7. Multivariate Approach

The next analysis we performed using AI aims to select a subset of inputs (same clinicopathological parameters as before) with the maximum relevance to the outputs (Claudin-7 expression). A wrapper feature selection method uses a classification algorithm to assess the influence of increasingly larger subsets of input attributes over the output. This approach is a greedy one: starting with a single input, it successively adds other inputs and measures the resulting classification accuracy. The greedy approach implies that there is no backtracking and no exhaustive attempt to assess all possible attribute combinations, which would quickly become intractable for a medium to a large number of inputs.

## 3. Results

### 3.1. Claudin-7 Expression

In this retrospective study, the immunohistochemical assessment of Claudin-7 expression was performed on 84 tumor samples, with histologically confirmed advanced stage CRC (stage IV). At the time of the last clinical follow-up, 64 patients (76.2%) out of the total group had died. Expression of Claudin-7 was defined as the presence of membranous staining in tumor cells. As shown in Table 1, the proportion of stained tumor cells was graded 3 for almost all cases, both in the core and invasive front. However, Claudin-7 staining intensity showed a larger variation in the scoring system, both in the core and invasive front. In the control group samples, represented by normal colonic mucosa, the intensity and the proportion of the Claudin-7 immunoexpression were both graded as 3. 

A discontinuous membranous staining pattern of Claudin-7 was observed mostly in the tumor invasion front (Table 2).

Claudin-7 immunoexpression in the normal colonic mucosa, staining intensity variations according to the scoring system and different Claudin-7 staining patterns (discontinuous vs. continuous) are shown in Figure 3.

#### 3.1.1. Clinicopathological Parameters and Claudin-7 Expression

In order to evaluate the relation between Claudin-7 expression (in the tumor core, respectively in the tumor invasive front) and clinicopathological parameters, correlations were made using Chi-squared and Fisher’s exact test (Table 3). A significant correlation (*p* = 0.033) was identified between Claudin-7 invasive front intensity and tumor leukocyte infiltrate, implying that a decrease in Claudin-7 intensity in the tumor invasive front is associated with an increase in leukocyte infiltrate. However, Claudin-7 expression regarding the proportion and intensity was not correlated with age, sex, T stage, N stage, grading, tumor location, venous invasion, lympho-vascular invasion, perineural invasion, growth pattern, tumor deposits and tumor budding (*p* > 0.05).

#### 3.1.2. Survival and Claudin-7 Expression

As previously mentioned, survival was defined as the time elapsed between the date of diagnosis and therapy initiation, to the date of death or of the last follow-up.

The Kaplan–Meier univariate survival analysis using the log-rank test showed a significant correlation between survival and Claudin-7 intensity in the invasive front (*p* = 0.00), with a higher expression (score 3) being associated with a worse prognosis. This decrease in survival occurred independently of Claudin-7 intensity in the tumor core, which had no impact on survival. In addition, scatter plots were also used for the statistical analysis of survival data (Figure 4).

### 3.2. Claudin-7 vs. Survival Rate AI Analysis

Table 4 and Table 5 present the results of the algorithms mentioned in the Materials and Methods section for the two considered problems (classification and regression, respectively), in terms of accuracy for classification and coefficient of determination for regression. The algorithms in Table 4 and Table 5 are different because some algorithms cannot be applied for both types of problems.

#### 3.2.1. Explicit Rules with Frequent Support

Some rules generated with the NNGE algorithm are presented below. These rules were detected from the whole dataset used for training. In the brackets at the end of each rule, the number of instances covered by the rule is included. Given the findings from the previous sections, it must be stressed that they may not be statistically relevant from the generalization point of view, but only a frequent pattern in the data. Still, these rules show some information related to higher rates of survival and they may need to be checked by the medical experts in case they provide some helpful clues regarding the appropriate treatment:class 5 IF: Cldn7_Core_P in {3} and Cldn7_Core_I in {2} and Cldn7_Core_D in {c} and Cldn7_Front_P in {3} and Cldn7_Front_I in {2} and Cldn7_Front_D in {c,d} (4)class 5 IF: Cldn7_Core_P in {3} and Cldn7_Core_I in {2,3} and Cldn7_Core_D in {c,d} and Cldn7_Front_P in {2} and Cldn7_Front_I in {1,2} and Cldn7_Front_D in {d} (4)class T IF: Cldn7_Core_P in {3} and Cldn7_Core_I in {2,3} and Cldn7_Core_D in {c} and Cldn7_Front_P in {1,3} and Cldn7_Front_I in {1} and Cldn7_Front_D in {d} (4)class T IF: Cldn7_Core_P in {3} and Cldn7_Core_I in {3} and Cldn7_Core_D in {c} and Cldn7_Front_P in {3} and Cldn7_Front_I in {2} and Cldn7_Front_D in {d} (6)


The following rules consider the six Claudin-7 attributes as inputs, together with the age of the patient in years:class 5 IF: Cldn7_Core_P in {3} and Cldn7_Core_I in {3} and Cldn7_Core_D in {c} and Cldn7_Front_P in {3} and Cldn7_Front_I in {1,2,3} and Cldn7_Front_D in {d} and 62.25 <= Age <= 81.74 (12)class 5 IF: Cldn7_Core_P in {3} and Cldn7_Core_I in {2} and Cldn7_Core_D in {c} and Cldn7_Front_P in {3} and Cldn7_Front_I in {1,2} and Cldn7_Front_D in {d} and 60.27 <= Age <= 67.05 (4)class 5 IF: Cldn7_Core_P in {3} and Cldn7_Core_I in {2} and Cldn7_Core_D in {c,d} and Cldn7_Front_P in {2,3} and Cldn7_Front_I in {1,2} and Cldn7_Front_D in {c,d} and 72.73 <= Age <= 76.67 (4)class T IF: Cldn7_Core_P in {3} and Cldn7_Core_I in {3} and Cldn7_Core_D in {c,d} and Cldn7_Front_P in {3} and Cldn7_Front_I in {2} and Cldn7_Front_D in {d} and 52.0 <= Age <= 56.41 (4)


#### 3.2.2. Correspondence between Clinicopathological Parameters and Claudin-7

For our case study, Table 6 shows the *InfoGain* measure for each combination of inputs and outputs.

#### 3.2.3. Multivariate Approach

In the following experiments, we use the order of attributes found by information gain and use the C4.5 decision tree algorithm, which naturally uses information gain for node splits. The full training set is used for these case studies. The results of this technique are presented in Figure 5.

For output O1, only I4 and I8 are enough to correctly classify the instances (Figure 5). This is not the case for the other outputs, which require all the inputs for maximum accuracy. For example, in the case of output O2, Cldn7 (Core) I, if only input I3, T-stage, is used, the decision tree has an accuracy of 60.5263%. If input I8, Leukocyte Infiltrate, is added, i.e., only the attributes I3 and I8 are used, the classification has a larger accuracy of 69.7368%. When I3, I8, and I4 are used, the accuracy becomes 80.2632%. Adding more inputs no longer increases the accuracy until the final attribute I11 is added, which leads to a 100% classification accuracy. The other subfigures in Figure 5 present the increase in accuracy for different orders of inputs corresponding to their respective outputs, as found by the information gain method.

## 4. Discussion

This retrospective study was designed to investigate the immunohistochemical expression of Claudin-7 and the potential prognostic significance in advanced CRCs. Furthermore, we aimed to draw a parallel between classical statistical algorithms and those used in AI.

Since its first discovery, a variety of studies suggested a link between Claudin-7 low expression and CRCs development and progression [5,27,28,29,30,31,32]. However, other studies obtained opposing results [9,33].

The present study identified a significant correlation between Claudin-7 intensity in the tumor invasive front and tumor leukocyte infiltrate (*p* = 0.033), implying that a decrease in Claudin-7 intensity in the tumor invasive front is associated with an increase in leukocyte infiltrate. To our knowledge, this is the first study to investigate the correlation of Claudin-7 immunohistochemical expression with inflammatory infiltrate in CRCs in humans. Consistent with our findings, but in animal models, Wang et al. reported that loss of Claudin-7 increases colonic infiltration of leukocytes during experimental colitis and demonstrated the promotion of colitis and associated CRC colitis in a Claudin-7 knockout mouse model [34].

We found that, in almost all cases (98.8% in the tumor core and 91.66% in the tumor invasive front), more than 60% of the tumor cells (grade 3) showed a high expression of Claudin-7. Similar results were reported by Kuhn et al. and Darido et al. where a low expression of Claudin-7 was found in normal colonic crypts, while in CRCs the expression was high. They identified the Claudin-7-EpCAM-CO-029-CDn44v6 complex and its upregulation also in hepatic metastasis of patients with CRC and a significant correlation was found between this complex and the clinical diversity, apoptosis resistance, and disease-free survival [9,33].

On the other hand, in contradiction to our results and also to the above-mentioned studies, Bornholdt et al. found an intense immunohistochemical reaction of Claudin-7 in normal colonic tissue, but a decreased or absent reaction in dysplastic and CRC tissue. These observations were sustained by the Claudin-7 mRNA levels, suggesting an early change in CRC carcinogenesis [29]. Moreover, Xu et al. proved that the positivity rate of Claudin-7 expression was significantly lower in CRC tissues than in peritumoral normal tissue and Claudin-7 expression was correlated with the grade of differentiation, being downregulated in well-differentiated adenocarcinomas, further downregulated in moderately differentiated adenocarcinomas, and significantly downregulated in poorly differentiated adenocarcinoma [5].

When analyzing the relationship between Claudin-7 expression and the morphological manifestations of EMT (tumor budding) we found no significant correlation. However, in contrast to our results Philip et al., Bhat et al. and Wang et al. concluded that Claudin-7 low expression or downregulation induced EMT, which plays a major role in CRC invasion, progression and metastasis process [30,31,32]. Furthermore, Xu et al. analyzed the effects of Claudin-7 knockdown in CRC stem cells through cell proliferation assay, migration assay, apoptosis assay and reported changes in cell characteristics such as promotion of cell proliferation, migration, and inhibition of cell apoptosis and the presence of EMT [35].

The present study found that when using a statistical approach, except for the leukocyte inflammatory infiltrate, Claudin-7 expression was not correlated with other clinicopathological parameters: age, sex, T stage, N stage, grading, tumor location, venous invasion, lympho-vascular invasion, perineural invasion, growth pattern, tumor deposits and tumor budding (*p* > 0.05), nor in the tumor core, neither in the tumor invasive front). Consistent with our results, Hou et al. who conducted a study to explore the role of Claudin-7, a p53 regulated gene, in tumorigenesis and progression of CRC through quantitative real-time PCR, Western blot, a luciferase reporter assay, and immunohistochemistry, found no correlation between clinicopathological parameters (tumor size, invasion depth, lymphatic metastasis, stage III/IV) and Claudin-7 high expression. In addition, Claudin-7’s high expression was significantly correlated with a favorable prognosis [36]. However, in contrast with these findings, in our study, the Kaplan–Meier univariate survival analysis, by using the log-rank test and the scatter plots, showed a significant correlation between survival and Claudin-7 intensity in the invasive front (*p* = 0.00), where a higher expression was associated with a worse prognosis.

Whenever conducting multivariate analysis, such as with the Cox regression method, one of the major pitfalls is having an insufficient number of outcome events (such as death) relative to the number of variables analyzed in the model. This proportion has been termed EPV (events per variable) [37,38], and a small value of the EPV affects the accuracy (risk estimates) and precision (95% confidence intervals) of odds or hazard ratios of the variables included, which may render misleading results. An adequate minimum value of the EPV is 10–20 (at the very least 10 outcome events per variable analyzed) [37]. In keeping with this rule, we found multivariate analysis unsuitable for our study (84 cases, 64 events, 15 variables).

Taken together, our study advocates for the potential prognostic and therapeutic role of Claudin-7 in advanced CRCs.

Concerning the study of AI, in this paper, we presented various analyses based on machine learning methods. In concordance with the findings of the statistical approach, we found that none of the applied algorithms can properly predict the survival rate based on the Claudin-7 inputs, therefore it is likely that there is no correlation between these inputs and outputs (Table 4 and Table 5). We have considered the input data both in discrete and continuous forms.

When considering the influence of clinicopathological parameters on Claudin-7 expression based on information gain, by analyzing the inputs on the rows in Table 6, one can identify the most relevant inputs for all the outputs. The applied information gain algorithm found that the most relevant single inputs are Leukocyte Infiltrate and N-stage. Age was found to be relevant for the Cldn7 (Core) P, Cldn7 (Front) P and Cldn7 (Front) I. T-stage was found to be relevant only for the Cldn7 (Core) I expression. These findings are consistent only with the correlation found by statistical means between Leukocyte Infiltrate and Claudin-7 intensity in the tumor invasive front.

The analysis based on subsets of input attributes shows that no single attribute can lead to a high accuracy classification. The actual accuracy values depend on the base classification algorithm wrapped for feature selection. For the multivariate approach using AI algorithms, we used C4.5 decision trees and information gain algorithms, which generally provide good results for the analyzed issue.

We found that two or three attributes from the set of Leukocyte Infiltrate, T-stage, N-stage, and Age increase the accuracy to more than 75% for each of the four outputs (Cldn7 (Core) P, Cldn7 (Core) I, Cldn7 (Front) P, and Cldn7 (Front) I). For Cldn7 (Core) P and Cldn7 (Front) P, the results are very good, with accuracies of 100% and 92%, respectively. However, with the exception of Cldn7 (Core) P, all input attributes are necessary to obtain maximum accuracy.

Even if the accuracy remains constant when adding more attributes, this does not mean that they are irrelevant to the classification. Figure 6 presents a similar evolution of accuracy for O2, but only considers the attributes that cause an increase in accuracy in Figure 5. It can be observed that only an accuracy of 82.8947% is eventually attained for these six inputs.

Regarding the fact that our method is not able to fully classify output O4 in Figure 5, we must stress the fact that the final accuracy depends not only on the subset of attributes but also on the classification algorithm used by the wrapper. In this case, C4.5 is not able to perform very well; however, other algorithms may have different levels of performance. The same wrapper procedure using a Random Forest leads to another evolution of accuracy, which finally reaches 100%, as shown in Figure 7.

## 5. Conclusions

Given the fact that the patients’ database is reduced (under 100 patients), classical machine learning methods seem to be a reasonable choice. With various error rates, all the applied algorithms support the finding that the survival rate cannot be predicted based only on Claudin-7 expression, yet there are correlations between clinicopathological parameters and Claudin-7, e.g., Leukocyte Infiltrate. Different outcomes might result from applying more complex deep learning techniques, but they may require a larger database of patients, since less data may result in overfitting and thus unreliable results. However, classical statistical algorithms have once again proven their crucial role in this research field.

By contrast, from a statistical point of view, the study showed that immunohistochemical intensity overexpression of Claudin-7 in the tumor invasive front may represent a poor prognostic factor in the advanced stages of CRCs.

## Figures and Tables

**Figure 1 cancers-14-02915-f001:**
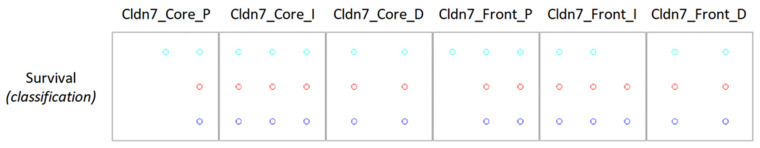
Visualization of independent inputs vs. outputs, for the classification problem (Cldn7—Claudin-7; P—percentage, I—intensity, D—membranous staining pattern: discontinuous vs. continuous).

**Figure 2 cancers-14-02915-f002:**
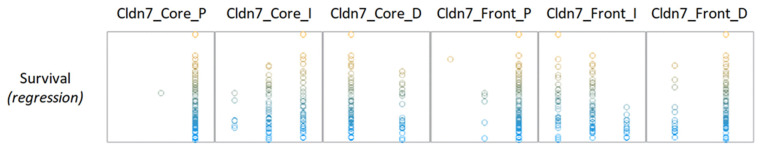
Visualization of independent inputs vs. outputs, for the regression problem (Cldn7—Claudin-7; P—percentage, I—intensity, D—membranous staining pattern: discontinuous vs. continuous).

**Figure 3 cancers-14-02915-f003:**
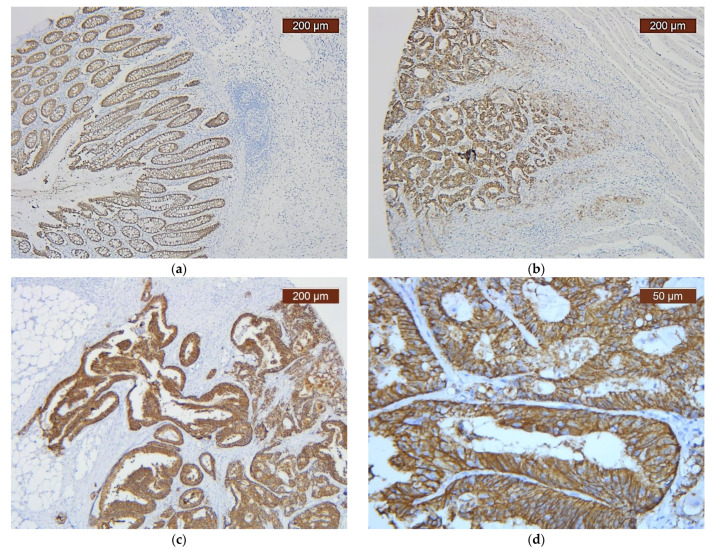
Immunohistochemical profile of Claudin-7 (IHC, anti-Claudin-7 monoclonal antibody) in: (**a**) normal colonic mucosa (magnification ×50); (**b**) CRC, intensity decreasing from core to invasive front (magnification ×50); (**c**) CRC, continuous membranous staining pattern in the tumor core and invasive front (magnification ×50); (**d**) CRC, continuous membranous staining pattern in the tumor core (magnification ×200); (**e**) CRC, discontinuous membranous staining pattern in the tumor core (magnification ×200); (**f**) CRC, discontinuous membranous staining pattern in invasive front (magnification ×100).

**Figure 4 cancers-14-02915-f004:**
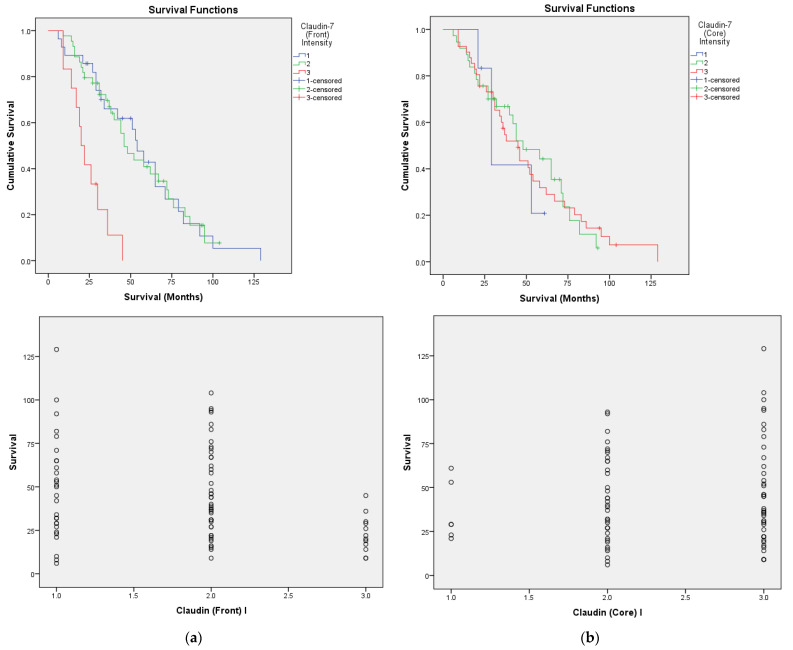
Prognostic significance of Claudin-7 intensity (I) expression regarding overall survival as calculated by Kaplan–Meier analysis and scatter plots: (**a**) invasive front; (**b**) tumor core. The intensity of immunoreaction: 1—weak, 2—moderate, 3—strong.

**Figure 5 cancers-14-02915-f005:**
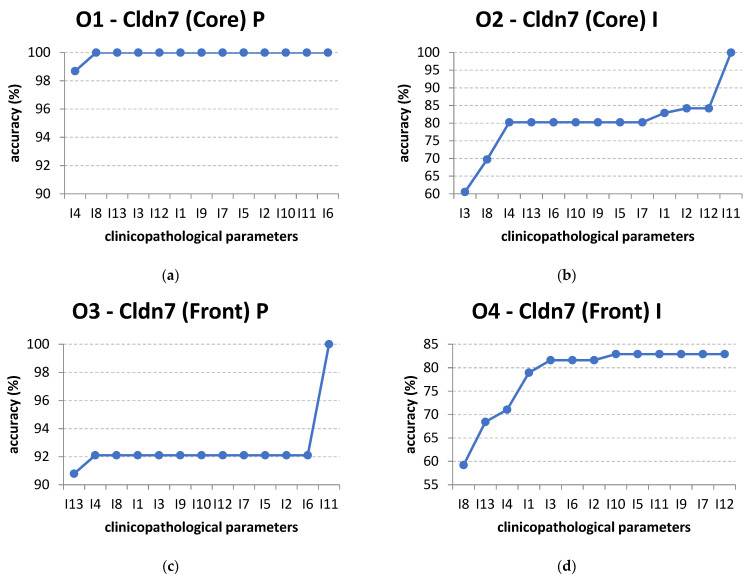
The increase of classification accuracy for each output by adding input attributes in the order specified by information gain: (**a**) output O1; (**b**) output O2; (**c**) output O3; (**d**) output O4.

**Figure 6 cancers-14-02915-f006:**
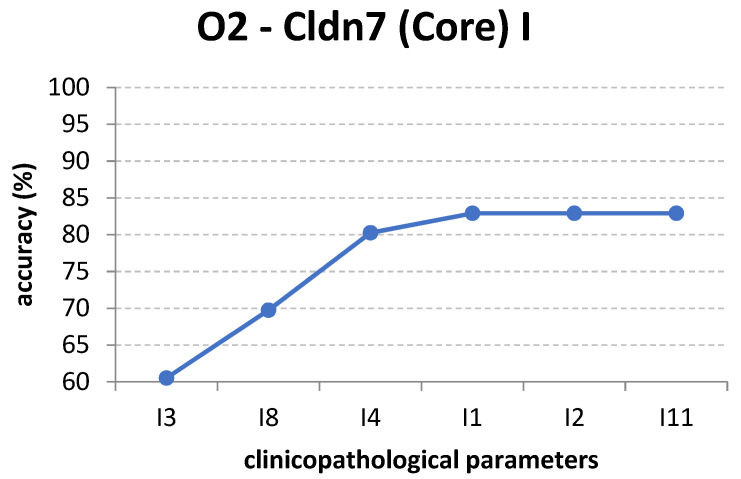
The increase of classification accuracy for output O2 for a bounded subset of inputs.

**Figure 7 cancers-14-02915-f007:**
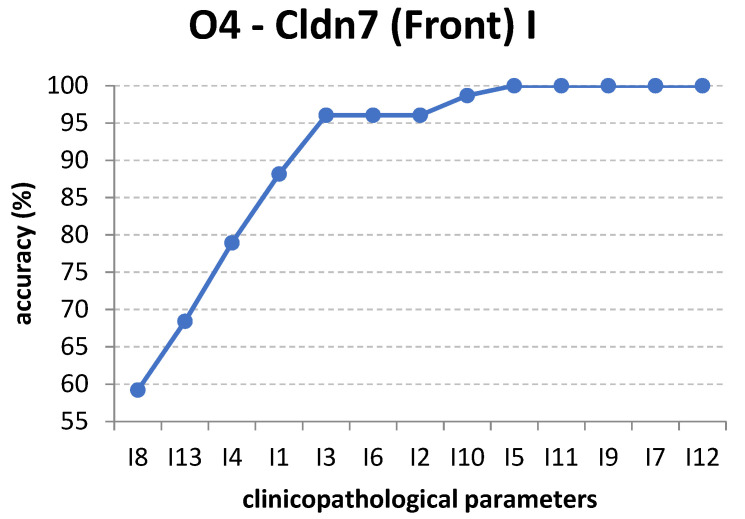
The increase of classification accuracy for output O4 using Random Forest.

**Table 1 cancers-14-02915-t001:** Comparison of Cldn7 expression between tumor core and invasive front.

	n	Min	Max	Mean
Cldn7 (Core) P	84	2	3	2.99
Cldn7 (Core) I	84	1	3	2.42
Cldn7 (Front) P	84	1	3	2.90
Cldn7 (Front) I	84	1	3	1.81
Control group	20	0	3	3

Abbreviations: Cldn7, Claudin-7; Cldn7 (Core) P, the proportion of Claudin-7 stained tumor cells in the tumor core; Cldn7 (Core) I, the intensity of Claudin-7 immunoreaction in the tumor core; Cldn7 (Front) P, the proportion of Claudin-7 stained tumor cells in invasive front; Cldn7 (Front) I, the intensity of Claudin-7 immunoreaction in invasive front.

**Table 2 cancers-14-02915-t002:** Comparison of Cldn7 staining pattern between tumor core and invasive front.

Staining Pattern	Cldn7 (Core)		Cldn7 (Front)	
	n	%	n	%
Continuous	62	73.8	15	17.9
Discontinuous	22	26.2	69	82.1
Total	84	100.0	84	100.0

Abbreviations: Cldn7, Claudin-7.

**Table 3 cancers-14-02915-t003:** Univariate analysis of Claudin-7 and clinicopathological parameters.

Clinicopathological Parameters		Cldn7 (Core) P		*p*-Value	Cldn7 (Core) I			*p*-Value	Cldn7 (Front) P			*p*-Value	Cldn7 (Front) I			*p*-Value
		2	3		1	2	3		1	2	3		1	2	3	
		*n*	*n*		*n*	*n*	*n*		*n*	*n*	*n*		*n*	*n*	*n*	
Age	21–30	0	1	0.310	0	1	0	0.655	0	0	1	0.227	0	1	0	0.647
	31–40	0	2		0	1	1		0	1	1		1	0	1	
	41–50	1	11		2	3	7		0	1	11		4	7	1	
	51–60	0	24		2	9	13		0	1	23		9	12	3	
	61–70	0	34		1	18	15		0	2	32		10	17	7	
	71–80	0	10		1	5	4		1	1	8		3	7	0	
	81–90	0	1		0	0	1		0	0	1		1	0	0	
Sex	F	1	35	0.429	3	16	17	1.000	1	3	32	0.523	13	18	5	0.953
	M	0	48		3	21	24		0	3	45		15	26	7	
T stage	T2	1	10		2	6	3		0	1	10		3	7	1	
	T3	0	41		3	21	17		0	4	37		17	17	7	
	T4a	0	27	0.190	1	8	18	0.206	1	1	25	0.744	8	15	4	0.389
	T4b	0	5		0	2	3		0	0	5		0	5	0	
N stage	N0	0	19		0	12	7		1	1	17		7	11	1	
	N1a	0	6		1	1	4		0	1	5		3	2	1	
	N1b	0	18		1	8	9		0	1	17		8	9	1	
	N1c	0	7	0.298	1	2	4	0.613	0	0	7	0.677	2	4	1	0.438
	N2a	1	11		1	5	6		0	2	10		4	7	1	
	N2b	0	22		2	9	11		0	1	21		4	11	7	
Grading	High Grade	0	15		1	6	8		0	1	14		3	8	4	
	Low Grade	1	68	0.821	5	31	33	0.906	1	5	63	1.000	25	36	8	0.278
Tumor Location	LeftColon	1	42		4	20	19		1	3	39		13	27	3	
	Rectum	0	10	1.000	0	6	4	0.625	0	2	8	0.312	6	2	2	0.053
	RightColon	0	31		2	11	18		0	1	30		9	15	7	
Venous Invasion	V0	0	20		0	12	8		0	1	19		5	10	5	
	V1	1	63	0.762	6	25	33	0.168	1	5	58	1.000	23	34	7	0.255
Lymphovascular Invasion	L0	0	9		0	7	2		0	1	8		3	6	0	
	L1	1	74	0.893	6	30	39	0.122	1	5	69	0.562	25	38	12	0.553
Perineural Invasion	Pn0	0	20		2	11	7		0	1	19		5	12	3	
	Pn1	1	63	0.762	4	26	34	0.352	1	5	58	1.000	23	32	9	0.640
Tumour Growth Pattern	Expansive	0	10		1	4	5		0	1	9		2	6	2	
	Infiltrative	1	73	0.881	5	33	36	0.882	1	5	68	0.603	26	38	10	0.651
Tumor Deposits	Absent	1	42		4	22	17		1	2	40		15	23	5	
	Present	0	41	0.512	2	15	24	0.244	0	4	37	0.427	13	21	7	0.832
Tumor Budding	Bd1	0	2		0	2	0		0	0	2		1	1	0	
	Bd2	0	12	1.000	1	7	4	0.354	0	2	10	0.438	3	6	3	0.743
	Bd3	1	69		5	28	37		1	4	65		24	37	9	
Leukocyte Infiltrate	3	0	1		0	1	0		0	0	1		0	1	0	
	5	0	23		1	10	12		0	2	21		5	15	3	
	10	0	26		3	10	13		1	3	22		13	11	2	
	15	1	8		2	2	5		0	1	8		5	2	2	
	20	0	11		0	5	6		0	0	11		3	5	3	
	25	0	1	0.143	0	0	1	0.366	0	0	1	0.876	1	0	0	0.033
	30	0	12		0	9	3		0	0	12		1	10	1	
	50	0	1		0	0	1		0	0	1		0	0	1	

Abbreviations: Cldn7, Claudin-7; Cldn7 (Core) P, the proportion of Claudin-7 stained tumor cells in the tumor core; Cldn7 (Core) I, the intensity of Claudin-7 immunoreaction in the tumor core; Cldn7 (Front) P, the proportion of Claudin-7 stained tumor cells in invasive front; Cldn7 (Front) I, the intensity of Claudin-7 immunoreaction in invasive front.

**Table 4 cancers-14-02915-t004:** The results of different algorithms for the classification problem.

Algorithm	Training	Cross Validation	Leave One Out
NN	60.7143%	36.9048%	40.4762%
kNN, k =10, w = 1/d	60.7143%	38.0952%	42.8571%
NNGE	48.8095%	45.2381%	42.8571%
C4.5	51.1905%	40.4762%	38.0952%
Random Forest, 100 trees	60.7143%	39.2857%	45.2381%
SVM, RBF kernel	60.7143%	34.5238%	39.2857%
Linear model	59.5238%	45.2381%	47.619%

**Table 5 cancers-14-02915-t005:** The results of different algorithms for the regression problem.

Algorithm	Training	Cross Validation	Leave One Out
NN	0.5803	0.3225	0.2759
kNN, k =10, w = 1/d	0.5801	0.2981	0.2524
Random Forest, 100 trees	0.5715	0.3069	0.2843
SVR, RBF kernel	0.55	0.156	0.1066
Linear regression	0.498	0.3268	0.2529

**Table 6 cancers-14-02915-t006:** The information gain values.

Clinicopathological Parameters		O1	O2	O3	O4
		Cldn7 (Core) P	Cldn7 (Core) I	Cldn7 (Front) P	Cldn7 (Front) I
I1	Tumor Location	0.01328	0.02698	0.03564	0.07025
I2	Grading	0.00389	0.00277	0.00406	0.02741
I3	T-stage	0.03746	** 0.14349 **	0.03254	0.05841
I4	N-stage	** 0.04147 **	0.10628	0.06594	0.10709
I5	Venous Invasion	0.00492	0.04001	0.00640	0.01225
I6	Lymphovascular Invasion	0.00241	0.05595	0.00358	0.03046
I7	Perineural Invasion	0.00525	0.02838	0.00738	0.00748
I8	Leukocyte Infiltrate	0.04120	0.13327	0.06169	** 0.17204 **
I9	Tumor Deposits	0.01328	0.04099	0.01975	0.01006
I10	Tumor Budding	0.00389	0.05448	0.01788	0.01834
I11	Tumor Growth Pattern	0.00274	0.00181	0.00324	0.01211
I12	Sex	0.01600	0.00256	0.01735	0.00179
I13	Age Group	0.03936	0.10040	** 0.06750 **	0.12619

Abbreviations: Cldn7—Claudin-7; P—percentage; I—intensity. In each column corresponding to an output Oi, the red values marked by bold typeface represent the first (i.e., best) value and the blue ones represent the second and third values.

## Data Availability

Not applicable.

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
