# Peer review of "Is High Expression of Claudin-7 in Advanced Colorectal Carcinoma Associated with a Poor Survival Rate? A Comparative Statistical and Artificial Intelligence Study"

_cancers, 2022, doi:10.3390/cancers14122915_

Round 1

Reviewer 1 Report

The article does not bring anything new as a scientific discovery but strengthens some already proven facts. It deserves to be published because it brings something new in the assessment given by artificial intelligence.

Author Response

Reviewer #1: The article does not bring anything new as a scientific discovery but strengthens some already proven facts. It deserves to be published because it brings something new in the assessment given by artificial intelligence.

We would like to thank reviewer 1 for his comments that have stressed on the positive aspects of our manuscript.

Reviewer 2 Report

In this manuscript entitled „Is High Expression of Claudin-7 in Advanced Colorectal Carcinoma Associated with a Poor Survival Rate? A Comparative Statistical and AI Study” Victor Ianole et al. focused on investigating the CRC tissue expression of Claudin-7 assessed by Tissue Microarray and IHC. The data are correlated with clinicopathological parameters and survival using paralel statistical and artificial intelligence (AI) algorithms. Claudin-7 is a tight junctions’ component maintaining the integrity of normal epithelial mucosa and is a potential prognostic factor in advanced colorectal cancer. An interesting finding that requires further investigation was that the Kaplan-Meyer univariate survival analysis showed a significant correlation between survival and Claudin-7 intensity in the invasive front and a higher expression was associated with a worse prognosis. In contrast, AI models couldn’t predict the same outcome on survival.  

Major comments:

The role of tight junction proteins in tumour progression and, as well as their implications in cancer metastasis is a very interesting topic of the last decade. Changes in the expression and distribution of tight junction proteins can result in loss in cohesion of the protein structure, which in turn results in the ability of cancer cells to become invasive. Claudin-7 role as a potential biomarker of CRC is clearly stated and supported through previous studies (Gowrikumar S et al., 2021; Xu C et al., 2021; Wang K et al., 2019; Li W et al., 2019, etc)    What is new, authors used AI study and compare with statistical result. The authors also shown the first time correlation of Claudin-7 expression with inflamatory infiltrate in CRC tissue. However, there are a few aspects to consider, as follows:

The authors only used one method for testing the presence of  Claudin-7 in CRC. It would be required to validate the IHC showing expression of Claudin-7 protein with Western blot method,

In my opinion, this evaluation should be done by two pathologists in blind analysis. The authors have done the analysis by two independent patologists in blind manner? It is also possible to use computer program to count the intensity of immunoreaction.

Have you checked the specificity of immunohistochemical staining by omitting the primary antibody and by replacing it with the same dilution of non-immunized serum?

The pictures are poor (yellow and overexposed background, different magnifications). Please remove into better quality ones.

Please add age, man/women to demographic and clinical-pathological features of colorectal cancer (CRC) patients.

Minor comments:

I would suggest adding hole names in title: artificial intelligence.

In discussion section remove lines 315-323 and add information to introduction section.

I would suggest adding graph as a scattergram to better depict the variability  immunoreactivity in poor prognosis patients.

Have you also check immunoexpression of Claudin-7 in adjacent normal tissue and in controls?

The authors should mention how many replicates were performed for the tissue staining images.

In introduction section add more information about claudin-7.

Author Response

Reviewers' comments

Reviewer #2: In this manuscript entitled „Is High Expression of Claudin-7 in Advanced Colorectal Carcinoma Associated with a Poor Survival Rate? A Comparative Statistical and AI Study” Victor Ianole et al. focused on investigating the CRC tissue expression of Claudin-7 assessed by Tissue Microarray and IHC. The data are correlated with clinicopathological parameters and survival using parallel statistical and artificial intelligence (AI) algorithms. Claudin-7 is a tight junctions’ component maintaining the integrity of normal epithelial mucosa and is a potential prognostic factor in advanced colorectal cancer. An interesting finding that requires further investigation was that the Kaplan-Meyer univariate survival analysis showed a significant correlation between survival and Claudin-7 intensity in the invasive front and a higher expression was associated with a worse prognosis. In contrast, AI models couldn’t predict the same outcome on survival.

Major comments:

The role of tight junction proteins in tumour progression and, as well as their implications in cancer metastasis is a very interesting topic of the last decade. Changes in the expression and distribution of tight junction proteins can result in loss in cohesion of the protein structure, which in turn results in the ability of cancer cells to become invasive. Claudin-7 role as a potential biomarker of CRC is clearly stated and supported through previous studies (Gowrikumar S et al., 2021; Xu C et al., 2021; Wang K et al., 2019; Li W et al., 2019, etc). What is new, authors used AI study and compare with statistical result. The authors also shown the first time correlation of Claudin-7 expression with inflamatory infiltrate in CRC tissue. However, there are a few aspects to consider, as follows:

1. The authors only used one method for testing the presence of Claudin-7 in CRC. It would be required to validate the IHC showing expression of Claudin-7 protein with Western blot method.

Thank you for this observation which is entirely correct. Unfortunately, due to the technical limitations we didn’t use a second method (e.g., Western blot) to sustain the results obtained using immunohistochemistry.

2. In my opinion, this evaluation should be done by two pathologists in blind analysis. The authors have done the analysis by two independent pathologists in blind manner? It is also possible to use computer program to count the intensity of immunoreaction.

Thank you for these questions. The evaluation was done by two pathologists in a blind manner. This statement is now included in the main text. At the present time, we do not have the possibility to realize computer assisted evaluation of the immunoreaction, although this would bring more accuracy and objectivity.

3. Have you checked the specificity of immunohistochemical staining by omitting the primary antibody and by replacing it with the same dilution of non-immunized serum?

Thank you for spotted this. We checked the specificity of the immunohistochemical staining in negative controls by omitting the primary antibody and by replacing it with non-immunized serum at the same dilution, for each TMA. This is now added to the main text.

4. The pictures are poor (yellow and overexposed background, different magnifications). Please remove into better quality ones.

Thank you for this observation. We replaced them with better pictures. We consider that presenting images at different magnifications can help the reader observe the details of the staining (at low magnification: core vs. front of invasion; at high magnification: intensity and continuity of the membrane staining). To be more relevant, we added picture with Claudin7 immunostaining in normal colonic mucosa from control group.

5. Please add age, man/women to demographic and clinical-pathological features of colorectal cancer (CRC) patients.

Thank you for this suggestion, we agree this is important. More than this, the AI analyzing methods revealed significant correlation between age and Claudin7 expression, while the classical statistical methods didn’t show significant correlations regarding these parameters. We added these new findings in the Results and Discussion subchapters.

Minor comments:

1. I would suggest adding whole names in title: artificial intelligence.

Thank you for this suggestion. The abbreviation was replaced.

2. In discussion section remove lines 315-323 and add information to introduction section.

Thank you for this suggestion, we agree. We moved the above mentioned paragraph from Discussion to Introduction.

3. I would suggest adding graph as a scattergram to better depict the variability immunoreactivity in poor prognosis patients.

Thank you for this suggestion. We added a Scatter diagram after the Kaplan-Meier plots. We considered that Kaplan-Meier plots are also useful to illustrate our results, because Claudin-7 was analyzed at a protein level, in a semi-quantitative manner (on a limited scale from 0 to 3).

4. Have you also check immunoexpression of Claudin-7 in adjacent normal tissue and in controls?

Thank you for this question. We agree the findings from normal tissue are important. To address this, we have included the evaluation in Paragraph 3.1, Figure 3 and Table 1.

5. The authors should mention how many replicates were performed for the tissue staining images.

Thank you for this suggestion. We added this information in Material and Methods.

6. In introduction section add more information about claudin-7.

Thank you for this suggestion. We added more information after rescanning the literature.

Reviewer 3 Report

This study reported Claudin-7 immunohistochemical expression was evaluated in tumor core and invasion front of advanced colorectal carcinoma (CRC) and correlated with clinicopathological parameters and survival using statistical and AI algorithms. The results are interesting and could have potential implications for CRC treatment. The materials were well-characterized, and the data were systematically analyzed. Conclusions from this work were supported by experiments or references. I don't really have technical comments, everything appears supported by data. However, there is one point that the authors might consider clarifying:

1)    Figure 4. The authors divide the CRC patients into three groups according to Claudin-7 immunohistochemical results. Please clarify how the three groups were differentiated based on the immunohistochemical scores.

Author Response

Reviewer #3: This study reported Claudin-7 immunohistochemical expression was evaluated in tumor core and invasion front of advanced colorectal carcinoma (CRC) and correlated with clinicopathological parameters and survival using statistical and AI algorithms. The results are interesting and could have potential implications for CRC treatment. The materials were well-characterized, and the data were systematically analyzed. Conclusions from this work were supported by experiments or references. I don't really have technical comments, everything appears supported by data. However, there is one point that the authors might consider clarifying:

  1. Figure 4. The authors divide the CRC patients into three groups according to Claudin-7 immunohistochemical results. Please clarify how the three groups were differentiated based on the immunohistochemical scores.

Thank you for the comments that have stressed on the positive aspects of our manuscript. Regarding the clarifications on the immunohistochemical scores, we consider them useful for the reader; therefore, we added explanations in the Figure 4 caption and in the preceding paragraph. Although we consider that Kaplan-Meier plots illustrate better our results, because Claudin-7 was analyzed at a protein level in a semi-quantitative manner (on a limited scale from 0 to 3), we added a Scatter diagram to differentiate the three groups.

Round 2

Reviewer 2 Report

The manuscript is ready to publish.